# Digestibility of black soldier fly larvae (*Hermetia illucens*) fed to leopard geckos (*Eublepharis macularius*)

**Kimberly L. Boykin**[1☉]*, **Renee T. Carter**[1‡], **Karina Butler-Perez**[1‡], **Cameron Q. Buck**[2‡],
**Jordan W. Peters**[1‡], **Kelly E. Rockwell**[3‡], **Mark A. Mitchell**[1☉]

**1** Department of Veterinary Clinical Sciences, School of Veterinary Medicine, Louisiana State University, Baton Rouge, LA, United States of America, **2** College of Veterinary Medicine, Tuskegee University, Tuskegee, AL, United States of America, **3** Berkeley Dog and Cat Hospital, Berkeley, CA, United States of America

☉ These authors contributed equally to this work.
‡ These authors also contributed equally to this work.
* kboykin@lsu.edu

**Data Availability Statement:** All relevant data are within the paper and its Supporting Information files.

## Abstract

Black soldier fly (BSF) larvae have been marketed as an excellent choice for providing calcium to reptiles without the need of dusting or gut loading. However, previous studies have indicated that they have limited calcium digestibility and are deficient in fat soluble vitamins (A, $D_3$, and E). In this feeding and digestibility trial, 24 adult male leopard geckos were fed one of three diets for 4 months: 1) whole, vitamin A gut loaded larvae; 2) needle pierced, vitamin A gut loaded larvae; or 3) whole, non-gut loaded larvae. Fecal output from the geckos was collected daily and apparent digestibility was calculated for dry matter, protein, fat, and minerals. There were no differences in digestibility coefficients among groups. Most nutrients were well digested by the leopard geckos when compared to previous studies, with the exception of calcium (digestibility co-efficient 43%), as the calcium-rich exoskeleton usually remained intact after passage through the GI tract. Biochemistry profiles revealed possible deficits occurring over time for calcium, sodium, and total protein. In regards to vitamin A digestibility, plasma and liver vitamin A concentrations were significantly higher in the supplemented groups (plasma- gut loaded groups: 33.38 ± 7.11 ng/ml, control group: 25.8 ± 6.72 ng/ml, t = 1.906, p = 0.04; liver- gut loaded groups: 28.67 ± 18.90 µg/g, control group: 14.13 ± 7.41 µg/g, t = 1.951, p = 0.03). While leopard geckos are able to digest most of the nutrients provided by BSF larvae, including those that have been gut loaded, more research needs to be performed to assess whether or not they provide adequate calcium in their non-supplemented form.

## Introduction

Black soldier fly (BSF) larvae (*Hermetia illucens*) are a popular feeder insect because they are the only commercially produced insect that has been found to potentially have a natural

**Funding:** Primary funding for this study was provided to KB and MM by the Louisiana State University VCS Corp grant (#8262). Additional support was provided by Fluker Farms, Inc. (flukerfarms.com) and Abaxis, Inc. (abaxis.com). The funder (Abaxis, Inc.) provided support in the form of research materials and did not have any role in the study design, data collection and analysis, decision to publish, or preparation of the manuscript. The funder (Fluker Farms, Inc.) provided support in the form of materials, funding, and salary to author [KB], but did not have any additional role in the study design, data collection and analysis, decision to publish, or preparation of the manuscript. The specific role of this author is articulated in the 'author contributions' section.

**Competing interests:** I have read the journal's policy and the authors of this manuscript have the following competing interests: KB is employed as a part-time dietary consultant at Fluker Farms, Inc. This does not alter our adherence to PLOS ONE policies on sharing data and materials.

positive calcium to phosphorous (Ca:P) ratio (2.5:1) based on thediet they are provided [1–5]. A positive calcium to phosphorous ratio (ideally 2:1 or greater) is considered essential for reducing the incidence of nutritional secondary hyperparathyroidism (NSHP) in insectivorous reptiles [6,7]. The black soldier fly larvae's high calcium level is due to an abundance of calcium carbonate impregnated within the exoskeleton [8]. However, previous studies and anecdotal reports have indicated that these larvae, in particular the calcium-rich exoskeleton, are poorly digestible and may not be capable of providing sufficient calcium to prevent disease. Dierenfeld and King (2008) looked at BSF larvae digestibility in mountain chicken frogs (*Leptodactylus fallux*) by analyzing fecal matter for nutrients and found that calcium digestibility was only 44% for frogs ingesting whole BSF larvae as compared to 84% when they were fed dusted crickets. When the BSF larvae (and exoskeleton) were mashed with a mortar and pestle, calcium digestibility increased to 88%. This suggests that physical disruption of the exoskeleton, such as would occur with mastication, could improve the calcium availability to the consumer. Dierenfeld and King theorized that species that were more apt to chew their food prior to ingestion would likely see higher levels of calcium digestibility [2]. Producers of the larvae have also made the recommendation that piercing the exoskeleton with a needle prior to feeding could help to improve issues with digestibility, but neither theory has been tested.

Of course, calcium is not the only nutrient that needs to be considered when constructing a diet for insectivorous species. In addition to NSHP, hypovitaminosis A is another common condition associated with insectivorous reptiles [9,10]. Most feeder insects, including BSF larvae, are severely deficient in the fat soluble vitamins (A, $D_3$, and E), and thus still require multi-vitamin dusting or gut loading [3,11–13]; however, recent research has found that some insects can produce vitamins $D_2$ and $D_3$ secondary to ultraviolet B radiation exposure, similar to vertebrates [14]. Vitamin A is integral to many bodily functions, including growth and development, immunity, vision, reproduction, and health and function of glands, ducts, and mucous membranes [10, 15–17]. Although true requirements for vitamins are not known for any reptile species, generic recommendations are currently extrapolated from requirements of laboratory rats [12, 18]. Previous research from the authors has proved the feasibility of gut loading vitamin A into BSF larvae [1], but determination of reptile digestibility and absorption requires further study.

The goal of this research was to determine if BSF larvae, given their potential issues with digestibility and vitamin deficiencies, can provide an adequate source of nutrition for leopard geckos (*Eublepharis macularius*). Leopard geckos were selected as a model because they are a common insectivorous species in the pet trade with well-documented nutritional disorders such as NSHP and hypovitaminosis A [6,9]. Additionally, their smaller size makes them more apt to at least partially chew their food as compared to larger species. Our specific objectives were to determine digestibility of whole versus needle-pierced BSF larvae in a leopard gecko model and to determine if calcium and vitamin A can be absorbed in sufficient quantities by the target species without the use of dusting powder supplementation. We hypothesized that 1) digestibility of BSF larvae would increase by piercing the exoskeleton with a needle prior to being offered to the leopard geckos, and that 2) geckos receiving vitamin A gut loaded BSF larvae would have higher liver vitamin A concentrations compared with those receiving non-gut loaded BSF larvae.

## Materials and methods

### Animals and husbandry

This research was approved and conducted in accordance with the rules and regulations set by the Louisiana State University's Animal Care and Use Committee (protocol #17–083).

Twenty-four male leopard geckos of unknown age were obtained from a private breeding colony with an average initial weight of 50.7 ± 11.5 grams (range 31.6–72.1 g). Males were recruited to rule out any potential bias associated with female reproduction. Each gecko was individually housed in a clear plastic terrarium (43 cm x 21 cm x 25 cm) without substrate and maintained at 28–30˚C (83–86˚F) and 30–40% relative humidity. Overhead fluorescent lamps provided lighting on a 12:12 light:dark cycle. The lighting did not provide any ultraviolet B radiation. The geckos had access to hide houses and water *ad libitum*. The baseline diet fed during the acclimation period was comprised of fasted crickets (4-week-old nymphs from Fluker Farms, Inc., Port Allen, LA) and BSF larvae (size large from Fluker Farms) at 3% of the gecko's body weight. Vitamin A concentrations in these insects were well under 100 μg/kg (as fed), which is considered low. Twenty of the geckos were maintained on this diet for 75 days before starting the feed trial, while four geckos underwent a 26 day acclimation period with the latter group needed to replace four original subjects that refused to eat BSF larvae.

## Physical and ophthalmic exams

Prior to the start of the feeding experiments, each gecko underwent a full physical exam. Fecal samples were collected and no parasites were seen on fecal floatations using zinc sulfate solution or direct 0.9% saline smears. Repeat physical exams were performed at the end of the study. In order to assess the geckos for ocular changes associated with potential hypovitaminosis A [19], ophthalmic exams were performed on awake geckos by a boarded veterinary ophthalmologist (RTC) at the beginning and end of the study. Exams included full visual assessment of the external structures, cornea, anterior chamber, and lens by slit-lamp biomicroscopy and evaluation of intraocular pressure (IOP) by rebound tonometry (Icare® Tonovet, Vantaa, Finland).

## Sedation and bloodwork values

Each animal was sedated to facilitate handling for physical exams and venipuncture. Sedation was achieved by using a combination of dexmedetomidine (Dexdomitor, 0.1 mg/ml; Zoetis Services LLC, Parsippany, NJ) at 0.1 mg/kg and midazolam (1 mg/ml; West-Ward Pharmaceuticals Corp., Eatontown, NJ) at 1 mg/kg, subcutaneously in the axillae [20,21]. Once the examinations were completed, the geckos were reversed with atipamezole (Antisedan, 5 mg/ml; Zoetis Services LLC, Parsippany, NJ) at 1 mg/kg and flumazenil (0.1 mg/ml, manufacturer) at 0.05 mg/kg, subcutaneously in the axillae. Due to issues with recovery in another leopard gecko study running simultaneously, the dexmedetomidine dose was decreased to 0.05 mg/kg and flumazenil was discontinued for all sedation events performed after the baseline blood draws. Sedation level remained appropriate for venipuncture with this dexmedetomidine dose.

Blood was collected via the cranial vena cava using a 3 ml syringe and 25-gauge needle. Maximum volume that could be safely drawn was 1.5% body weight (0.5–1.1 ml). Samples were placed into lithium heparin microtainers, centrifuged within 30 minutes of collection, and chilled on ice. Within three hours of collection, all plasma was separated and frozen at -80˚C until further processing could be performed. Two blood samples were obtained during the acclimation/vitamin A depletion period. On Day -50, blood was collected for a baseline plasma vitamin A concentration (measured as retinol, n = 13). These samples were sent to Michigan State University (MSU Veterinary Diagnostic Laboratory, Lansing, MI) for evaluation using ultra high performance liquid chromatography (UPLC). A second sample, obtained on Day -35 (n = 20), was used for an in-house plasma biochemistry panel (VetScan VS1 Chemistry Analyzer, Abaxis, Inc., Union City, CA). Two separate sampling periods were required due to the blood volume required for each test (>0.25 ml for UPLC, >0.1 ml for

biochemistries) and difficulties in drawing enough blood at one time. For the four replacement geckos, a baseline vitamin A plasma concentration from MSU was not performed and the samples for the biochemistry profile were obtained on Day -14 (n = 4). An additional blood sample was drawn during the course of the study on Day 35 for a repeat plasma biochemistry profile (n = 23). On Days 105 and 140, samples from individual geckos were pooled together for a final plasma vitamin A concentration using UPLC from MSU (n = 24 and 24, respectively).

## Feeding experiments

The 24 geckos were divided into three diet groups using a random number generator (random.org). The treatment groups received either vitamin A gut loaded BSF larvae that were intact (Group 1, n = 8) or pierced once with a 21-gauge needle (Group 2, n = 8). The control group (Group 3, n = 8) received only non-gut loaded BSF larvae that remained intact. BSF larvae were gut loaded for 24 hours using a wheat bran and corn meal based diet with an added water-soluble vitamin A supplement (Rovimix-A 500-WS [178,500 μg retinyl acetate/g, equal to 150,000 μg retinol equivalents/g]; DSM Nutritional Products, Ames, IA). The expected final concentration of vitamin A in the diet was 20,000 μg/kg (dry matter basis, DMB) and the final larval vitamin A concentration was expected to be 1,000 μg/kg (as fed basis or 3,636 μg/kg DMB) [1]. Over the course of the study, samples of the wheat bran diet (n = 4) and larvae (treatment groups, n = 8; control, n = 8) were sent to SDK Laboratories, Inc. (Hutchison, KS) for vitamin A concentration analysis performed by high performance liquid chromatography (HPLC). All geckos were offered larvae equal to 5% of their body weight three times per week. The weights of BSF larvae ingested by the geckos were recorded over the course of the study (140 days) to calculate total vitamin A ingestion.

## Apparent digestibility of BSF larvae

In order to measure BSF larvae digestibility, fecal samples produced during the first and last month of the study were collected and pooled together by month and feeding group (n = 2 for each group). Any water dishes that contained feces were transferred to a glass dish and dried in an oven at 100˚C for several hours before being combined with the rest of the fecal material. Fecal samples were frozen at -80˚C until they could be further processed. Nutrient content analysis was performed by Dairy One Forage Lab (Ithaca, NY). Nutrient analysis was also performed on representative samples of BSF larvae (gut loaded and control). Average daily intake was determined for each group of geckos and apparent nutrient digestibility was calculated based on the dry matter intake and excretion of each nutrient using the following formula:

$$Apparent\ digestibility\ (\%) = \frac{Average\ daily\ intake - Average\ daily\ output}{Average\ daily\ intake} \times 100$$

## Liver biopsies

At the conclusion of the study, all geckos underwent anesthesia and surgical liver biopsies for determination of liver vitamin A concentrations. Geckos were fasted for 12 hours prior to premedication with dexmedetomidine (0.1 mg/kg), midazolam (1 mg/kg), and hydromorphone (2 mg/ml, Hospira, Inc., Lake Forest, IL) at 0.25 mg/kg subcutaneously in the axillae. The geckos were maintained on isoflurane inhalant gas via face mask during the procedure. Anesthesia was monitored throughout the procedure via respiratory rate, Doppler heart rate, and muscle tone. The surgical site was prepared aseptically with chlorhexidine scrub 2% and sterile 0.9% saline. A #11 scalpel blade was used to make a 1–1.5 cm left paramedian incision starting 1–1.5 cm caudal to the xiphoid process. Both lobes of the liver were visually assessed for any

abnormalities. The left lobe was then gently exteriorized using sterile cotton tipped applicators and digital manipulation. The caudal half of the left lobe was clamped using Hemoclips (blue or medium size, Weck Hemoclip® Traditional; Teleflex Medical, Research Triangle Park, NC) and removed with a scalpel blade, placed in a Whirl-Pak® bag (Nasco, Fort Atkinson, WI), and frozen at -20˚C until processed. The surgical site was closed using a 2-layer method and 4–0 Vicryl suture (polyglactin 910, Ethicon US, LLC, Somerville, NJ); the body wall was closed in a simple continuous pattern and the skin was closed using a horizontal mattress pattern.

The geckos were reversed with atipamezole following the same protocol noted previously. Meloxicam (OstiLox ™, VetOne, Boise, ID) at 0.2 mg/kg subcutaneously was also given post-operatively. A second dose of hydromorphone (0.25 mg/kg) was given 18–24 hours after surgery, and meloxicam at 0.2 mg/kg subcutaneously was continued once per day for four days. Seven geckos (Group 1, n = 3; Group 2, n = 2; Group 3, n = 2) were euthanized intra-operatively after collection of the biopsies due to marked weight loss toward the end of the study period. All seven were submitted for full necropsy.

## Statistical analysis

Sample size for this study was determined using the following a priori information: an alpha = 0.05, a power = 0.80, an expected difference in vitamin A liver concentrations of 20 μg/g, and a standard deviation for the treatment and control groups of 12 μg/g. The Shapiro-Wilk test, skewness, kurtosis, and q-q plots were used to evaluate the distributions of the data. Data that were normally distributed are reported as mean and standard deviation (SD), while non-normally distributed data are reported as median and 25–75 percentiles (%), and min-max. Non-normal data were log transformed for parametric testing. Outliers were identified using the Dixon's Q test and removed if the calculated Q value was larger than the critical value given a 95% confidence interval. The amount of vitamin A ingested per group was dependent on voluntary gecko intake, therefore, larvae intake was analyzed for significance among groups using a one-way ANOVA. Fecal digestibility coefficients were also analyzed using a one-way ANOVA. Paired plasma biochemistry data was analyzed using a repeated measures ANOVA. None of the ANOVA tests required post-hoc analysis to differentiate significance between diet groups. For final vitamin A concentrations in the plasma and liver, Groups 1 and 2 were combined into a single vitamin A gut loaded group versus the non-gut loaded control group (Group 3). The two groups were then analyzed using a one-tailed independent t-test. A Pearson's correlation test was used to determine if there was a relationship between the presence of a mucoid ocular discharge and the liver or plasma vitamin A concentrations. A commercial software (SPSS 25.0; IBM Statistics, Armonk, NY) was used to analyze the data; $p < 0.05$ was used to determine statistical significance.

## Results

### Physical exams and necropsy results

All geckos were determined to be healthy at the beginning of the study. Over the course of the experiment, 10 (41.6%) out of 24 leopard geckos lost weight (mean ± SD: -13.76 ± 7.71% weight loss, range: -0.14% to -23.91%)(Group 1, n = 4; Group 2, n = 3; Group 3, n = 3). Seven (29.2%) of these geckos experienced inappetence and weight loss severe enough to require early removal from the study (Group 1, n = 3; Group 2, n = 2; Group 3, n = 2). These geckos were euthanized intra-operatively (after collection of hepatic biopsies) and then submitted for necropsy. Necropsy results revealed stomatitis (n = 4 total; mild to moderate, n = 3; moderate to marked, n = 1), hepatic lipidosis/vacuoles (n = 4), infection with *Cryptosporidium* spp. via histopathology

and/or fecal floatation (n = 3), lymphoplasmacytic enteritis (n = 3), renal tubular necrosis (n = 2), and pulmonary xanthomatosis (n = 1). None of the necropsied specimens had evidence of epithelial squamous metaplasia that could be associated with hypovitaminosis A.

## Ophthalmic exams

No obvious ophthalmic lesions were observed at the beginning of the study. Exams were repeated at the end of the study. Seven (29.2%) of the geckos did have mild mucoid discharge within the medial canthus of one or both eyes (Group 1, n = 2; Group 2, n = 2; Group 3, n = 3). Three of these animals (Group 1, n = 1; Group 2, n = 1; Group 3, n = 1) were among those that were necropsied and showed no evidence of squamous metaplasia or any other ocular disease. Correlation analysis showed no significant correlation between the presence of the discharge and the liver (r = 0.371, p = 0.074, two-tailed) or plasma (r = -0.47, p = 0.105, two-tailed) vitamin A concentrations. The authors are unsure as to the significance of the mucoid discharge, as no other significant ocular disease process was found.

## Diet vitamin A concentrations and intake amounts

The target vitamin A concentrations for the larval diet and larvae were 20,000 µg/kg (DMB) and 1,000 µg/kg (as fed or 3,636 µg/kg DM), respectively. When analyzed samples were averaged together over the course of the experiment, the vitamin A supplemented diet contained 32,753 ± 15,350 µg/kg (DMB) and the larvae for the two treatment groups analyzed at 835 ± 232 µg/kg (as fed or 3,036 ± 844 µg/kg DMB). The average vitamin A concentration of the control group diet and larvae were 391 ± 40 µg/kg (DMB) and 23 ± 30 µg/kg (as fed or 84 ± 109 µg/kg DMB), respectively. Geckos were allowed to eat the larvae *ad libitum* up to 5% of their body weight. On average, the geckos ate 54.7 ± 12.2% of the food offered to them per feeding (2.74% body weight per feeding). When intake was analyzed as the average amount of grams eaten per group per feeding, there was no significant difference among diet groups (F = 0.587, p = 0.558).

## Apparent digestibility of BSF larvae

The average nutritional composition of BSF larvae used in this study is presented in the first column of Table 1. The mean digestibility coefficient ± standard deviation for the pooled feces from each group is listed in the remaining columns of Table 1. There was no significant difference in digestibility among groups for any of the nutrients analyzed. Since there were no significant differences among groups, all digestibility coefficients were averaged together and reported as a single mean ± standard deviation in the first column of Table 2. The second and third columns of Table 2 show the reported values for BSF larvae digestibility from the mountain chicken frog study [2]. With the exception of calcium and phosphorous, leopard geckos were more capable of digesting whole BSF larvae than mountain chicken frogs.

## Blood values

Baseline plasma vitamin A (measured as retinol) concentrations were all found to be <50 ng/ml (lower limit of quantification for small volume samples, <1 ml). For the final plasma vitamin A samples, only 13 out of 24 samples were of a high enough volume to obtain readings above the limit of quantification. One sample in the non-gut loaded group was returned as <20 ng/ml, and for purposes of including for statistical analysis, was read as 20 ng/ml. For the 13 samples that were able to be analyzed, vitamin A concentrations were significantly higher

**Table 1. Nutritional composition of BSF larvae and apparent digestibility coefficients for leopard geckos fed an exclusive diet of BSF larvae prepared in one of three ways.**

| Nutrient | Average BSF larvae composition | Group 1: Intact, vitamin A gut loaded (n = 2) | Group 2: Pierced, vitamin A gut loaded (n = 2) | Group 3: Intact, non-gut loaded (n = 2) |
|---|---|---|---|---|
| Dry Matter, % | 27.4 ± 0.81 | 70 ± 0.9 | 72 ± 9.3 | 71 ± 4.5 |
| Crude Protein, % | 56.1 ± 0.91 | 81 ± 0.5 | 82 ± 3.8 | 80 ± 1.7 |
| Crude Fat, % | 23.2 ± 11.78 | 65 ± 11.7 | 74 ± 13.7 | 67 ± 5.9 |
| Ash, % | 9.85 ± 0.33 | 49 ± 9.3 | 54 ± 11.3 | 54 ± 6.1 |
| Calcium, % | 2.14 ± 0.08 | 41 ± 8.9 | 42 ± 18.1 | 44 ± 6.1 |
| Phosphorous, % | 1.15 ± 0.03 | 42 ± 4.1 | 47 ± 10.8 | 46 ± 3.2 |
| Magnesium, % | 0.39 ± 0.02 | 50 ± 5.8 | 52 ± 8.8 | 52 ± 3.0 |
| Potassium, % | 1.35 ± 0.03 | 77 ± 1.4 | 81 ± 2.9 | 80 ± 3.4 |
| Sodium, % | 0.13 ± 0.01 | 62 ± 3.9 | 65 ± 13.8 | 17 ± 90.9 |
| Iron, mg/kg | 204 ± 7.72 | 58 ± 4.9 | 60 ± 2.6 | 58 ± 2.8 |
| Zinc, mg/kg | 131 ± 5.24 | 37 ± 5.4 | 39 ± 1.0 | 43 ± 2.5 |
| Copper, mg/kg | 11.2 ± 0.38 | 26 ± 6.0 | 31 ± 4.4 | 33 ± 0.3 |
| Manganese, mg/kg | 232 ± 8.94 | 24 ± 8.1 | 20 ± 15.9 | 27 ± 5.1 |
| Molybdenum, mg/kg | 1.24 ± 0.08 | 44 ± 8.5 | 49 ± 10.9 | 52 ± 2.1 |
| Sulfur, % | 27.4 ± 0.8 | 41 ± 0.7 | 47 ± 9.0 | 45 ± 4.5 |

No significant difference was found among groups for any nutrient.

(t = 1.906, p = 0.0415) in the treated groups (n = 8; 33.38 ± 7.11 ng/ml) compared with the control group (n = 5; 25.8 ± 6.72 ng/ml) (Table 3).

Plasma biochemistries were sampled on Day -35 (baseline) and Day 35. A final sample was not performed due to the plasma volume restrictions of the vitamin A test. No significant differences were seen among groups during either of the two time periods; however, when baseline values (averaged across all three groups) were compared to Day 35 values using a repeated measures ANOVA, there were significant decreases in calcium, total protein, albumin,

**Table 2. Apparent digestibility coefficients for leopard geckos fed a long term (140 days) exclusive diet of BSF larvae compared to published digestibility coefficients for mountain chicken frogs (*Leptodactylus fallux*) fed either whole or mashed BSF larvae for a period of only one day each.**

| Nutrient | Leopard Geckos, whole and pierced larvae (n = 6) | Frogs, Whole Larvae[*] (n = 5) | Frogs, Mashed Larvae[*] (n = 5) |
|---|---|---|---|
| Dry Matter, % | 71 ± 4.7 | 26 ± 9.9 | 76 ± 3.2 |
| Crude Protein, % | 81 ± 2.0 | 44 ± 7.5 | 77 ± 3.1 |
| Calcium, % | 43 ± 9.5[a] | 44 ± 7.5[a] | 88 ± 1.7 |
| Phosphorus, % | 45 ± 5.8[a] | 51 ± 6.5[a] | 91 ± 1.3 |
| Magnesium, % | 51 ± 5.0 | 6.3 ± 13 | 40 ± 8.0 |
| Potassium, % | 79 ± 2.7 | 24 ± 10 | 60 ± 5.3 |
| Sodium, % | 48 ± 48 | -378 ± 64 | -489 ± 78 |
| Copper, mg/kg | 30 ± 4.6 | -61 ± 21 | 50 ± 6.7 |
| Iron, mg/kg | 59 ± 2.9 | -284 ± 51 | 40 ± 7.9 |
| Molybdenum, mg/kg | 49 ± 7.2 | -126 ± 56 | 23 ± 9.6 |
| Zinc, mg/kg | 40 ± 3.8 | 23 ± 10 | 72 ± 3.7 |

[*]Data reported in study by Dierenfeld and King, 2008

[a] The digestibility coefficients for calcium and phosphorous were the only nutrients to not differ significantly when comparing our data to the values reported for intact larvae from Dierenfeld and King (2008) using a single value t-test

**Table 3. Plasma and liver vitamin A concentrations from geckos receiving vitamin A gut loaded BSF larvae versus those receiving non-gut loaded BSF larvae.**

|  | Vitamin A supplemented (Groups 1 & 2) | Non-gut loaded (Group 3) |
|---|---|---|
| Final Plasma Vitamin A concentrations (retinol, ng/ml) | 33.38 ± 7.11 (n = 8) | 25.80 ± 6.72* (n = 5) |
| Final Liver Vitamin A concentrations (total vitamin A, µg/g) | 28.67 ± 18.90 (n = 16) | 14.13 ± 7.41** (n = 7) |

All samples were run by Michigan State University's Veterinary Diagnostic Laboratory using UPLC. Eleven plasma samples were not included due to insufficient plasma quantities reading below the limit of quantification for this assay (LOQ: <50 ng/ml).

*One sample was reported as <20 ng/ml. This value was included as a reading equal to 20 ng/ml.

**A single outlier of 61.35 µg/g was removed from this group prior to statistical analysis.

globulin, and sodium (Table 4). One value for calcium during the baseline time period was >16 mg/dL (upper limit of quantification for the rotors used) and one value for albumin at the Day 35 time period was <1 g/dL (lower limit of quantification for the rotors used). For statistical purposes these values were removed, however, their removal did not change the significance found for these analytes. One value from potassium at Day 35 was also removed for obvious hemolysis interference. Additionally, bile acids were under 35 umol/L for all individuals tested with the exception of one individual at the Day 35 time period that was reported as 44 umol/L. Bile acids are not reported in Table 4.

## Liver biopsies and surgical outcomes

Liver biopsies collected at the end of the study revealed a significantly higher hepatic vitamin A concentration (measured as total vitamin A) for the gut loaded group versus the control group (Vitamin A supplemented: 28.67 ± 18.90 µg/g, Non-gut loaded: 14.13 ± 7.41 µg/g; t = 1.951, p = 0.0325)(Table 3). A single outlier was removed from the control group (liver vitamin A concentration of 61.35 µg/g). Age, genetic factors, or diet prior to the study could have played a role in the unusually high liver concentration of this gecko compared to the rest of the geckos in this group. There were no anesthetic complications with any of the surgeries. All non-survival surgeries (n = 7) were determined prior to the start of anesthesia. All survival surgeries (n = 17) were performed successfully with only one animal requiring the placement of Gelfoam® Sterile Sponge (Pfizer Inc., New York, NY) for more controlled hemostasis. Post-

**Table 4. Average biochemistry results from all geckos over time (baseline vs. day 35).**

| Biochemistry Analytes | Baseline Values | Day 35 Values | Significance |
|---|---|---|---|
| AST (U/L) | 54.6 ± 20.3 | 67.9 ± 37.2 | F = 2.314, p = 0.144 |
| Creatinine kinase (U/L) | 458 (25% = 275, 75% = 1006) | 698 (25% = 286, 75% = 1256) | F = 0.877, p = 0.361 |
| Uric acid (mg/dL) | 3.4 ± 1.5 | 3.8 ± 1.4 | F = 1.359, p = 0.257 |
| Glucose (mg/dL) | 166.5 ± 14.4 | 161.6 ± 20.1 | F = 1.207, p = 0.285 |
| Calcium (mg/dL) | 14.2 ± 1.1 | 13.0 ± 1.1 | F = 25.299, p<0.001* |
| Phosphorous (mg/dL) | 3.8 ± 0.6 | 3.6 ± 0.8 | F = 1.38, p = 0.254 |
| Total Protein (g/dL) | 5.9 ± 0.9 | 5.1 ± 0.7 | F = 19.061, p<0.001* |
| Albumin (g/dL) | 1.8 ± 0.3 | 1.5 ± 0.3 | F = 30.076, p<0.001* |
| Globulin (g/dL) | 4.1 ± 0.7 | 3.7 ± 0.5 | F = 10.717, p = 0.004* |
| Potassium (mmol/L) | 5.2 ± 0.6 | 5.0 ± 0.8 | F = 1.993, p = 0.174 |
| Sodium (mmol/L) | 147.8 ± 5.8 | 138.8 ± 3.1 | F = 94.955, p<0.001* |

No significance was found between groups at either time period. All data was normally distributed with the exception of creatinine kinase which was able to be log transformed for analysis. Mean ± standard deviation is reported for normally distributed data. Median and 25–75 percentiles (%) are reported for creatinine kinase. F statistics and p values are reported for all analytes; those with an asterisk (*) indicate significance, p<0.05.

operatively, there were no issues with hemorrhage, dehiscence, or any other wound complications. As of one-year post-surgery, no additional issues or concerns were noted.

## Discussion

The results of this study confirm that leopard geckos are capable of digesting BSF larvae. Overall, BSF digestibility was higher in this reptile species compared to a single study with an amphibian [2]. Average dry matter digestibility of intact BSF larvae was 71 ± 2.8% when fed to leopard geckos and 26 ± 9.9% for mountain chicken frogs. Protein, magnesium, potassium, sodium, iron, zinc, copper, and molybdenum also saw significant gains in digestibility (Table 2). Some of the differences in digestibility could be related to differences in larval composition between the studies. However, given that most of our digestibility coefficients were more similar to the values reported for mashed larvae from their study (Table 2, column 3) rather than the values for intact larvae (Table 2, column 2), we believe that most of the differences were due to a higher degree of mastication by the leopard geckos which would allow for digestive enzymes to breach the tough exoskeleton and breakdown the inner portions of the larvae. This would also explain why piercing the BSF larvae with a needle did not result in improved digestibility among the three groups (Table 1), as the geckos were already accomplishing this through mastication. It is possible that needle-piercing would still be of benefit to species that swallow their prey whole. Transecting the BSF larvae could also be considered, but this can lead to decreased movement by the larvae and reduced acceptance rate by the geckos (personal observation).

Calcium digestibility, on the other hand, did not improve and was similar between the two species (43% for geckos, 44% for frogs) when fed intact BSF larvae [2]. It would appear that gecko mastication or needle piercing does not provide enough disruption to the exoskeletal matrix to allow for calcium carbonate digestion. Visual assessment of the geckos' feces supports this hypothesis, as there would frequently be whole exoskeletons passed through the digestive tract (personal observation). This poor level of calcium digestibility should raise concerns over whether non-supplemented BSF larvae can support the calcium needs of captive reptiles. The estimated minimum dietary calcium requirement for growing leopard geckos is between 6.1 and 8.5 g Ca/kg diet (DMB) [22]. When adjusted for digestibility, BSF larvae in our study provided 9.2 g Ca/kg diet (DMB), which should be adequate to support calcium needs. It is possible that the calcium digestibility was low in these leopard geckos because they were adults and calcium absorption was impacted by normal feedback mechanisms. A study in fast-growing juvenile geckos would be helpful in further discerning whether low digestibility is a function of the BSF exoskeleton, as it would be expected to be higher in growing animals with a higher calcium requirement.

However, the authors would still recommend caution due to the fact that physiological needs may vary based on species, age, reproductive status, and/or vitamin D status of the animal. Additionally, calcium content of BSF larvae can vary greatly based on their rearing diet's composition. Recent studies have shown calcium:phosphorous ratios can range anywhere between 0.3:1 to 14.9:1 [4,5]. Thus, more research would be needed to prove that non-supplemented BSF larvae can, in fact, provide enough calcium to insectivorous reptiles based on how the insects are reared.

When comparing the paired blood samples, there was a significant decrease in calcium over time for all treatment groups. Poor calcium digestibility or low levels of vitamin D could be possible causes for this decline. Another explanation could be that the geckos received high levels of calcium supplementation prior to entering the study and that the decline represented a return to more physiologic levels. Unfortunately, due to plasma volume restrictions at the

conclusion of the study, a final biochemistry panel to determine whether or not values continued to trend down could not be performed. As all of the geckos were males, reproductive status did not play a role in this decline. At this time, we do not have enough data to recommend whether calcium supplementation is needed when offering BSF larvae to reptiles. Additional research needs to be conducted to establish true calcium requirements for reptiles and varied diets should always be offered to insectivores to limit the incidence of nutritional deficiencies.

The biochemistry panels also revealed significant decreases in total protein, albumin, globulins, and sodium in the geckos over time for all treatment groups. Compared to other feeder insects, BSF larvae have lower concentrations of protein and sodium [3,11]. The decrease in protein could indicate a change from a primarily protein-rich cricket diet to one of a higher fat, lower protein larval diet. Hyporexia could also play a role in the decreased values, but with an average intake of 2.7% body weight and no correlation between blood values and the amount of food ingested, this is less likely. In regards to sodium, there is added concern about the possibility of poor or negative digestibility. Negative digestibility indicates that more sodium is being lost in the feces than was initially present in the food. This was seen in both mountain chicken frogs and corn snakes (*Pantherophis guttatus*) fed diets containing BSF larvae [2,23]. It was also seen in a single replicate from Group 3 of this study. Dierenfeld and King originally theorized that the BSF larvae may be irritating the gut and causing a "diarrhea"-type syndrome that could lead to hyponatremia if fed exclusively over time [2]. Unfortunately, no other studies have addressed digestibility of sodium or looked at sodium biochemistry data from animals ingesting BSF larvae. To the authors' knowledge, the present study is the first to report serum or plasma sodium concentrations for any animal consuming BSF larvae as the main ingredient of the diet. When compared to reference intervals for this species, the sodium levels obtained from Day 35 of this study do indicate a possible hyponatremia, however, these reference intervals may not be representative of the true physiologic range for sodium in leopard geckos [24]. Feeding a variety of insects or protein sources, should alleviate any concerns that could be associated with the nutritional deficits of an exclusive BSF larval diet.

The second objective regarding digestion and absorption of vitamin A from gut loaded larvae was more challenging to confirm. Previous research in multiple animal species suggested that plasma vitamin A concentrations are not usually representative of whole body vitamin A status, nor do they correlate well with liver vitamin A concentrations [25–27]. Retinol-binding proteins within the blood tend to maintain homeostatic concentrations of circulating vitamin A, except in cases of extreme hypo- or hypervitaminosis A [16, 28]. Liver concentrations are considered the more accurate representation of body status, but clinically can be more challenging to obtain in sick or small patients. Ideally, liver biopsies from each gecko would have been obtained at the beginning and end of the study period. Without baseline hepatic concentrations of vitamin A for individual geckos, it is impossible to truly evaluate a change in vitamin A status, but random assignment of the geckos to treatment groups should have minimized any bias. Unfortunately, due to the geckos' small size and the minimum requirements for biopsy weight (0.25 grams), only one hepatic biopsy would be able to be performed unless the study became non-survival. The other issue was the possibility of fatal complications early on in the study period leading to reduction in sample population sizes. A recent study reported a high complication rate following liver biopsies in this species, so the authors had concerns about multiple surgeries [29]. Due to these limitations and the fact that there are no published references for plasma vitamin A concentrations in leopard geckos, the authors decided to proceed with collecting plasma concentrations as well.

The minimum plasma volume required for the vitamin A assay used in this study was 0.15 ml; however, the limit of quantification for sample volumes below 1 ml is 50 ng/ml. For

samples that are larger than 1ml, the limit of quantification can go as low as 10 ng/ml. Unfortunately, due to size restrictions, our baseline sample volumes all fell below 1 ml of plasma and all results were returned as <50 ng/ml. Previous literature has reported plasma vitamin A (measured as retinol) concentrations for various squamates and amphibian species, including green iguanas (*Iguana iguana*, 52–75 ng/ml), eastern indigo snakes (*Drymarchon couperi*, 9 ng/ml), anacondas (*Eunectes murinus*, 80 ng/ml), Mississippi gopher frogs (*Rana capito servosa*, 36–43 ng/ml), marine toads (*Bufo marinus*, 60 ng/ml), Cuban tree frogs (*Osteopilus septentrionalis* 83 ng/ml), and Puerto Rican crested toads (*Bufo lemur*, 130 ng/ml) [25–27, 30–33]. To the authors' knowledge, there is no other literature reporting plasma vitamin A concentrations in leopard geckos, or any other insectivorous lizard, so these baseline values either indicate that leopard geckos have lower than average circulating plasma vitamin A concentrations compared to other squamates and insectivorous amphibians or that these geckos were already deficient prior to being included in this study. Given that all individuals were deemed healthy at the beginning of the study, and that no geckos developed lesions that would be considered classical for diagnosis of hypovitaminosis A (epithelial squamous metaplasia), the former is probably more likely.

Only 13 of the 24 final plasma samples were reported back with values reading below 50 ng/ml (Group 1, n = 7; Group 2, n = 1; Group 3, n = 5). Thus, for comparative analysis between treatments, data from Groups 1 and 2 were combined, providing a single vitamin A gut loaded treatment group versus the control (Group 3). Final plasma concentrations did show a significant difference between groups (t = 1.906, p = 0.0415), with the vitamin A supplemented geckos having higher plasma vitamin A concentrations. Liver vitamin A concentrations also differed significantly between the vitamin A gut loaded groups and the control group (t = 1.951, p = 0.0325). Across both treatment groups, liver vitamin A concentrations ranged from 2.9–77.98 μg/g, with only a few individuals near the upper end of this range. These geckos likely had higher liver concentrations at the start of the study compared to the others. Age, diet, and husbandry conditions prior to the study (all of which are unknown), are likely contributors to the wide range seen. Paired liver samples would have helped limit this variance, but due to the limitations already discussed, were not performed.

One other study has measured liver vitamin A concentrations in leopard geckos. Cojean *et al.* (2018) used female leopard geckos to determine if differences existed between liver vitamin A uptake and storage when geckos were given pre-formed vitamin A supplementation versus beta-carotene [29]. Similar to the present study, Cojean's group also only performed biopsies at the end of the study and did not collect paired samples. Those authors found that the beta-carotene fed group displayed higher overall liver concentrations of vitamin A compared with the pre-formed vitamin A group, despite the theory that many carnivorous/omnivorous reptiles are not capable of converting beta-carotene to the active form (mean 13.43 μg/g, min-max 2.31–24.05 μg/g vs. mean 9.49 μg/g, min-max 6.76–13.33 μg/g, respectively). The values from the present study are much higher, with differences in sex (all female vs. all male study designs) and age being major influencing variables. Females would potentially have lower body stores of vitamin A due to large quantities being stored in the developing eggs. Cojean's geckos were also 6–9 months old compared to ours that were mostly thought to be adults (> 10 months old). Vitamin A tends to accumulate in the liver as an animal ages leading to potential differences in the two populations." [34].

In contrast to Cojean's study, we experienced no major surgical or anesthetic complications. Seventeen geckos underwent a successful liver biopsy of the left lobe (median weight = 0.22g, min = 0.08g, max = 0.59g). Lower overall vitamin A stores or some other aspect of folliculogenesis may have contributed to poor wound healing and dehiscence. Additionally, different anesthetic protocols and biopsy techniques (guillotine vs. hemoclip) may have led to

better success in the present study. The authors would strongly encourage further studies involving liver biopsies on leopard geckos in the future, if they are indicated. One such study would be to determine plasma and liver vitamin A concentrations in wild-caught leopard geckos. This much needed study would help researchers to determine if our various methods of supplementation in captivity are meeting the physiological needs of this species. As far as which supplementation (beta-carotene or pre-formed vitamin A) is better for leopard geckos, there is still some debate, but both supplements appear to have some degree of absorbance and assimilation into the gecko and should probably be used in combination along with other carotenoid sources.

Without supplementation with either pre-formed vitamin A or appropriate precursors, insectivorous reptiles are at a high risk for developing hypovitaminosis A. The lesions most often associated with this disease are hyperkeratosis and epithelial squamous metaplasia, with the most classical presentations manifesting as ocular lesions (periocular edema, conjunctivitis, blepharitis, ocular discharge and debris) [9,10,15]. Wiggans *et al*. (2018) reported that 46% of all leopard geckos that visited a veterinary teaching hospital between 1985 and 2013 had some form of ocular lesion. Lack of vitamin A in the diet was one of the major risk factors associated with development of ocular disease [19]. In the present study, mild mucoid discharge was seen in seven geckos across all treatments and was not correlated with plasma or liver vitamin A concentrations.

Other symptoms may vary by species and have included nasal discharge, gular edema, swollen or thickened lips, stomatitis, vertebral kinking, hemipenal impaction, and renal tubular hyperkeratosis leading to visceral gout [10,15,17,35]. The timeline for developing these lesions is still currently unknown. Kroenlein *et al*. (2008) found no difference in liver vitamin A concentrations nor histological evidence of squamous metaplasia in red-eared sliders (*Trachemys scripta elegans*) after six months of vitamin A depletion. Research in humans and other adult vertebrates confirm that >6 months of depletion is usually needed before clinical signs of hypovitaminosis A are detectable, but will obviously depend on the vitamin A status of the individual prior to the start of depletion [36–38]. None of the seven geckos (Group 3, n = 2) that underwent necropsy had histopathological lesions consistent with hypovitaminosis A. A case could be made for stomatitis lesions being associated with hypovitaminosis A, but lesions were seen in treatment and control groups and were not correlated to vitamin A concentrations. Even though there was a significant difference seen between groups in regards to liver and plasma vitamin A concentrations, without histopathological evidence of disease, the authors are unable to conclude that the control group was or was not trending towards vitamin A depletion status. The difference in concentrations only provides evidence that gut loaded BSF larvae are capable of being digested enough to provide vitamin A to the consumer and to produce levels higher than geckos not receiving any supplementation.

Although BSF larvae appear to be well digested and able to offer gut loaded nutrients to leopard geckos, 10 out of the 24 individuals developed issues with inappetence and weight loss during the course of the study. The reason for this was likely multi-factorial, with a main issue being palatability. Even before the study started, four geckos had to be replaced due to strict refusal to eat BSF larvae. Most of the geckos that experienced weight loss during the study period would consistently eat less than the other geckos. Whether food preferences were due to palatability or a lack of movement of BSF larvae compared to crickets was outside the scope of this study. Another contributor to inappetence and weight loss was infection with *Cryptosporidium* sp. At least three geckos were confirmed to be infected at necropsy, but none showed obvious signs of infection until after the acclimation period had been completed. The stress of being transitioned to an exclusive diet of BSF larvae, instead of mixed feedings with crickets, may have increased the severity and progression of the disease. Knowing that the

prevalence of cryptosporidiosis is relatively high in the commercially bred population of leopard geckos, the authors were well aware that this parasitic disease could disrupt results for a digestibility study, but these animals ultimately represent what is available in the current pet trade. Fortunately, the geckos that were infected rarely ate well and, therefore, did not contribute much to the pooled feces. The other possible contributor that was found on necropsy was stomatitis. These lesions were not seen grossly, but inflammation was seen on histopathology. The cause of the stomatitis is unknown, but nutrient deficiencies or irritation from ingestion of the BSF larvae are possible. Once again, the authors strongly encourage feeding insectivores a wide variety of insects in order to provide the diverse nutrients they require.

## Conclusions

Overall, leopard geckos are able to digest intact BSF larvae better than previously reported for the mountain chicken frog. Despite high nutrient digestibility for proximate constituents, calcium digestibility remained low. While calcium levels were likely adequate for leopard geckos based on estimated calcium requirements for the species, further research is needed to verify this assumption and to determine calcium requirements for other insectivorous species. This study demonstrated that leopard geckos are able to utilize gut loaded nutrients such as vitamin A incorporated into feeder BSF larvae. Additional research is necessary to determine gut loading amounts for other nutrients (e.g., vitamins $D_3$ and E, etc.). Plasma and liver vitamin A concentrations for leopard geckos can be challenging to obtain in a clinical setting, but can be used in research to better study true requirements for this species.

## Supporting information

**S1 Appendix. Raw data for bloodwork, liver values, and digestibility analysis.**
(XLSX)

**S1 Checklist. ARRIVE guidelines checklist.**
(PDF)

## Acknowledgments

The authors would like to thank the Division of Laboratory Animal Medicine at LSU for assisting us with the care of these animals.

## Author Contributions

**Conceptualization:** Kimberly L. Boykin, Mark A. Mitchell.

**Data curation:** Kimberly L. Boykin, Mark A. Mitchell.

**Formal analysis:** Kimberly L. Boykin, Mark A. Mitchell.

**Funding acquisition:** Kimberly L. Boykin, Mark A. Mitchell.

**Investigation:** Kimberly L. Boykin, Renee T. Carter, Karina Butler-Perez, Cameron Q. Buck, Jordan W. Peters, Kelly E. Rockwell, Mark A. Mitchell.

**Methodology:** Kimberly L. Boykin, Renee T. Carter, Mark A. Mitchell.

**Project administration:** Kimberly L. Boykin, Mark A. Mitchell.

**Resources:** Kimberly L. Boykin, Mark A. Mitchell.

**Software:** Kimberly L. Boykin, Mark A. Mitchell.

**Supervision:** Kimberly L. Boykin, Mark A. Mitchell.

**Validation:** Kimberly L. Boykin, Mark A. Mitchell.

**Visualization:** Kimberly L. Boykin, Mark A. Mitchell.

**Writing – original draft:** Kimberly L. Boykin, Mark A. Mitchell.

**Writing – review & editing:** Kimberly L. Boykin, Renee T. Carter, Karina Butler-Perez, Cameron Q. Buck, Jordan W. Peters, Kelly E. Rockwell, Mark A. Mitchell.

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
