## [Decision Letter · Decision Letter 0]

6 Feb 2020

PONE-D-19-33399

Can reptiles digest black soldier fly larvae (*Hermetia illucens*)? Evaluating their digestibility using leopard geckos (*Eublepharis macularius*) as a model

PLOS ONE

Dear Dr. Boykin,

Thank you for submitting your manuscript to PLOS ONE. After careful consideration, we feel that it has merit but does not fully meet PLOS ONE’s publication criteria as it currently stands. Therefore, we invite you to submit a revised version of the manuscript that addresses the points raised during the review process.

Please address the comment by the reviewers to clarify the concerns brought up.

We would appreciate receiving your revised manuscript by Mar 22 2020 11:59PM. To enhance the reproducibility of your results, we recommend that if applicable you deposit your laboratory protocols in protocols.io, where a protocol can be assigned its own identifier (DOI) such that it can be cited independently in the future. For instructions see: http://journals.plos.org/plosone/s/submission-guidelines#loc-laboratory-protocols

We look forward to receiving your revised manuscript.

Kind regards,

Jake Kerby, Ph.D.

Academic Editor

PLOS ONE

Additional Editor Comments (if provided):

I was able to obtain three reviews for this manuscript and the reviewers had similar comments on many points. Please address these comments in a reply. Some are merely grammatical, but others are perhaps important to the overall conclusion of the paper.

Journal Requirements:

2. As part of your revision, please complete and submit a copy of the ARRIVE Guidelines checklist, a document that aims to improve experimental reporting and reproducibility of animal studies for purposes of post-publication data analysis and reproducibility: https://www.nc3rs.org.uk/arrive-guidelines. Please include your completed checklist as a Supporting Information file. Note that if your paper is accepted for publication, this checklist will be published as part of your article.

3. In your Methods section, please provide additional details regarding the BSF larvae used in your study and ensure you have described the source. For more information regarding PLOS' policy on materials sharing and reporting, see https://journals.plos.org/plosone/s/materials-and-software-sharing#loc-sharing-materials.

4. Thank you for providing the following Funding Statement: 

"Primary funding for this study was provided to KB and MM by the Louisiana State University VCS Corp grant (#8262). Additional support was provided by Fluker's Cricket Farm, Inc. (flukerfarms.com) and Abaxis, Inc. (abaxis.com). The funders had no role in study design, data collection and analysis, decision to publish, or preparation of the manuscript."

We note that one or more of the authors is affiliated with the funding organization, indicating the funder may have had some role in the design, data collection, analysis or preparation of your manuscript for publication; in other words, the funder played an indirect role through the participation of the co-authors.

If the funding organization did not play a role in the study design, data collection and analysis, decision to publish, or preparation of the manuscript and only provided financial support in the form of authors' salaries and/or research materials, please review your statements relating to the author contributions, and ensure you have specifically and accurately indicated the role(s) that these authors had in your study in the Author Contributions section of the online submission form. Please make any necessary amendments directly within this section of the online submission form.  Please also update your Funding Statement to include the following statement: “The funder provided support in the form of salaries for authors [insert relevant initials], but did not have any additional role in the study design, data collection and analysis, decision to publish, or preparation of the manuscript. The specific roles of these authors are articulated in the ‘author contributions’ section.”

If the funding organization did have an additional role, please state and explain that role within your Funding Statement.

Please also provide an updated Competing Interests Statement declaring this commercial affiliation along with any other relevant declarations relating to employment, consultancy, patents, products in development, or marketed products, etc. 

Reviewers' comments:

Reviewer's Responses to Questions

**Comments to the Author**

1. Is the manuscript technically sound, and do the data support the conclusions?

Reviewer #1: Partly

Reviewer #2: Partly

Reviewer #3: Yes

2. Has the statistical analysis been performed appropriately and rigorously? 

Reviewer #1: No

Reviewer #2: Yes

Reviewer #3: Yes

3. Have the authors made all data underlying the findings in their manuscript fully available?

Reviewer #1: Yes

Reviewer #2: Yes

Reviewer #3: Yes

4. Is the manuscript presented in an intelligible fashion and written in standard English?

Reviewer #1: Yes

Reviewer #2: Yes

Reviewer #3: Yes

5. Review Comments to the Author

Reviewer #1: Interesting study; specific comments appended in the MS.

In particular you need to pay detailed attention to the actual compound measured regarding vitamin A status - likely retinol - and associated units. Vitamin A, per se, is typically reported as IU or UI - based on the underlying compound measured. Especially with exotic species and the limited knowledge of nutrient requirements, physiology and metabolism - and particularly how they are evaluated and utilized, one must be explicit with defining actual compound(s) measured.

Reviewer #2: As detailed in my comments tot he authors my main concern is the conclusion that BSFL should be supplemented with calcium before being fed to leopard geckos. The data as presented does not seem to support that conclusion. Hence why I replied partly to question #1.

Reviewer #3: this is a well written manuscript and a well designed and important study. Some minor comments listed below.

Line 33-34: report p values as 0.04 and 0.03

Line 77: Please provide a reference for this statement (leopard geckos prone to vit a deficiency, etc).

Line 105: provide reference that leopard geckos develop ocular changes due to vitamin A deficiency

Line 191, 220, 230: the 7 emaciated geckos were from all 3 treatments groups? Would be worth reporting here.

Line 254-256: the statement regarding the frogs is inappropriate for the result section and should be moved to the discussion section.

Line 291: possible that this animal was a female, and not a male?

Line 323-324: were the larvae fed in your study and in the frog study of same size/age? Possible that older/larger larvae are harder to digest or have different nurtritional values than smaller/younger ones?

Line 350-359: Or it could be due to a vitamin D deficiency. What was the vit D3 content of the larvae. Did they receive enough vit D3? Since no UVB source was provided, dietary vit D3 content of the offered diet should be discussed here as a potential cause for lower Ca levels on day 35….

6. PLOS authors have the option to publish the peer review history of their article (what does this mean?). If published, this will include your full peer review and any attached files.

Reviewer #1: No

Reviewer #2: Yes: Mark D Finke

Reviewer #3: No

---

## [Author Response · Author response to Decision Letter 0]

1 Apr 2020

Dear academic editor and reviewers,

We would like to submit a revised version of the manuscript entitled “Digestibility of black soldier fly larvae (Hermetia illucens) fed to leopard geckos (Eublepharis macularius)" by KL Boykin, RT Carter, K Butler-Perez, CQ Buck, JW Peters, KE Rockwell, and MA Mitchell. Please see the following outline to review changes and comments that were made regarding the requested revisions. Line numbers correspond with the manuscript version (not the tracked changes version).

Sincerely,

Kim Boykin

Reviewer 1 comments:

Overall: In particular you need to pay detailed attention to the actual compound measured regarding vitamin A status - likely retinol - and associated units. Vitamin A, per se, is typically reported as IU or UI - based on the underlying compound measured. Especially with exotic species and the limited knowledge of nutrient requirements, physiology and metabolism - and particularly how they are evaluated and utilized, one must be explicit with defining actual compound(s) measured.

Comment to reviewer: I believe we have made the changes requested. Plasma vitamin A was measured as retinol and liver vitamin A was measured as total vitamin A. As for food, we wrote the units out in mcg/kg to follow convention with other papers that had been published this way, as well as due to the fact that IU seems to be on the way out. The new nutrition labels for the FDA require vitamins be listed in mcg or mg instead of IU.

Line 1: Changed title to be more succinct

Line 26: “between” changed to among

Live 39: “nutrition” and “reptiles” have been added as keywords

Line 67: changed “Vitamin A is important…” to “Vitamin A is integral to many different…”

Line 70: added the word generic

Line 70: changed “based on the known requirements” to “extrapolated from requirements…”

Line 72-73: changed “but determination of whether it can be digested and absorbed by a reptile” to “but determination of reptile digestibility and absorption requires further study.”

Line 77: added “in the pet trade,” to the sentence

Line 77: changed “that are well-known to be susceptible to nutritional disorders…” to “with well-documented nutritional disorders…”

Line 79: Is this true? Do they have teeth or palatine ridges? Suggest rephrase or omit.

Response to reviewer: Leopard geckos have pleurodont teeth that are replaced every couple of months (this has been well studied). I also have videos of them partially chewing the larvae prior to swallowing that I used to show at a conference where this research was originally presented. I’d be happy to email the video to you if you would like.

Line 82: changed “dusting powders” to “dusting powder supplementation”

Line 85: Changed “than” to “compared with”

Line 98-101: “Each gecko was started on a vitamin A depleted diet (fasted crickets and BSF larvae) during the acclimation period.” Did you analyze diet to confirm deficiency in vitamin A? Size and amounts of insects fed.

Changed to “The baseline diet fed during the acclimation period was comprised of fasted crickets (4 week old nymphs) and large BSF larvae at 3% of the gecko’s body weight. Vitamin A concentrations in these insects were well under 100 µg/kg (as fed), which is considered very low.”

Response to reviewer: More specific feeding amounts cannot be easily provided as we would only feed crickets periodically. They were usually only fed to those that were losing weight or not showing interest in eating BSF larvae for a week or more.

Line 102-103: “while four geckos underwent a 26 day acclimation period. The latter group was needed to replace four original subjects that would not eat BSF larvae.” Changed to “while four geckos underwent a 26 day acclimation period with the latter group needed to replace four original subjects that refused to eat BSF larvae.” 

Line 103: Comment- Palatability probably deserves a comment in discussion, almost 20% refusal is substantial.

 Comment to reviewer: Palatability is mentioned in discussion, line 508-513.

Line 109: Added the word potential “ocular changes associated with potential hypovitaminosis A”

Line 126: Add range for blood volumes drawn. Added (0.5-1.1 ml)

Line 132: Added n=13. We were not able to get blood out of everyone this round hence the decreased value. 

Line 131-132: Comment- Was this a pooled sample or from each individual?

Comment to reviewer: The first vitamin A sample was not pooled as we were unaware of the need to make it a pooled sample at this time. The second and final sample was pooled as was mentioned in the manuscript (line 142).

Line 132: Comment- As just retionol? Or other compounds and retionoids too? Need to be specific

 Comment to reviewer: Added “measured as retinol” to the sentence.

Line 132-133: We did not make the sentence reformatting change you suggested here because according to journal submission guidelines, Michigan State University has to be written out in its full form before it can be abbreviated and your suggested way would have needed parentheses inside of parentheses, making it look jumbled.

Line 134: Changed to “A second sample, obtained on Day -35 (n=20), was used…” 

Line 136-138: Two separate sampling periods were required “due to” the blood volume required for “each test (>0.25 ml for UPLC and >0.1 for biochemistries)” and difficulties in drawing “adequate quantities in a single sampling.”

Line 140: Added (n=4).

Line 141: Added (n=23).

Line 143: Added (n=24 and 24, respectively).

Line 145: Removed the word “randomly” 

Line 153: Comment- Provide this on a DM basis as well. This has been added (3,636 µg/kg DMB).

Line 157: Comment- Suggest you add a similar statement for the acclimation period. Did you not feed crickets during the trial period?

Comment to the reviewer: Geckos were fed 3% BW during the acclimation period but was bumped to 5% BW for the study because several were losing weight despite a good appetite. We don’t think it is necessary to the paper to add all of the details of acclimation and trial period as it would just continue to make the manuscript bulky. 

Crickets were occasionally fed during the study period (approximately 2-3 times per month) to ensure that some geckos would not lose too much weight. Feces were not collected near these days to ensure only BSF larvae were in the feces.

Line 157-159: Changed to “The weights of BSF larvae ingested by the geckos were recorded over the course of the study (140 days) to calculate total vitamin A ingestion.”

Line 163: Comment- If you actually did this, and not by individuals, then you cannot run statistics as you’d have n=1 only for each treatment group

Comment to reviewer: Although each group was pooled, we still had two batches from each pool that were able to be analyzed. We were supposed to have more, but we had issues with our lab animal workers cleaning cages and throwing away fecal samples when they weren’t supposed to.

Line 163: Changed to “Any water “dishes” that contained feces “were transferred to” a glass…”

Line 168: Comment: This needs better definition. Did you collect over different phases (i.e. 0-50 days, 51-100 days, 101-140 days, etc.) to get your subsamples? Unclear as currently written.

 Comment to reviewer: Although we did collect feces over the entire time period, we had issues with fecal samples being thrown away and leaving us with unusable data. We have clarified which days were actually used in our final analysis which ended up being the first and last month of the study. See line 161-162.

Line 203 and 204: changed “by the” to “as” 

Line 208: changed between to “among”

Line 207-209: Comment- You had 3 groups—did you also measure control?

Comment to reviewer: Wording seemed a little confusing. We calculated larvae intake (in grams) per group not the estimated vitamin A intake per group, sorry for any confusion. We changed it to say, “larvae intake was analyzed for significance among groups.” So yes, the control group was also calculated here.

Line 213: did not make suggested change (adding a hyphen between words “gut loaded”) as it does not match the format already established

Line 217: changed “data; p<0.05”

Line 224-228: Comment- Confusing- so only 4 of the 7 were replaced?

Comment to reviewer: Four geckos were removed before the study even started. These seven were removed approximately 1 month prior to the end of the study (day 140) due to inappetence and weight loss.

Line 239: Changed “Correlation analysis showed no significant…”

Line 244: Comment- Are you sure? This looks more like a target level in the diet of 20,000 IU/kg DM. If this is actually the target, conversion of µg retinol would equate to 66,667 IU/kg DM for the diet. Similar for the larvae- 1000 µg/kg would equate to 3333 IU/kg AFB in the larvae or close to 10,000 IU/kg DM.

Comment to reviewer: The values are correct as written. The larvae diet has to have an extremely high vitamin A content to be able to reach appropriate levels in the larvae. One of our cited papers (Finke 2003) can be used to confirm the high amounts needed for vitamin gut loading in insects.

Line 245: Comment- Convert this to DM as well. Probably something like 3000-3500.

 Comment to reviewer: Added “1000 µg/kg (as fed or 3,636 µg/kg DM)”

Line 246: Changed “was” to “contained”

Line 247: changed “were” to “analyzed at”

Line 247- 248: Did not accept change to combine the sentences because it makes the resulting sentence too long and it reads incorrectly.

Line 250: Comment- Ug is not a proper unit for reporting vitamin A per se- it is the unit for retinol and other retinoids. Convert to vitamin A in IU if you want to talk about that nutrient or keep using ug units referring to retinol and provide justification for your values. Conversions can be tricky but can be justified.

Comment to reviewer: As stated above, we reported food vitamin A in mcg/kg to follow another paper that was published regarding vitamin A gut loading of insects (Finke 2003) and because the FDA is starting to change nutrition labels to show mcg or mg instead of IU. The measurements we received back from the lab were in IU, but we converted them to this format because of the reasons outlined.

Line 253: Changed “between” to “among”

Line 256: Changed “no significant difference in digestibility among groups for any of the nutrients analyzed.”

Line 261: Remove the word “much”

Line 266: Changed “between” to “among”

Line 276: Changed “bloodwork values” to “blood values”

Line 277: Changed “Baseline plasma vitamin A (measured as retinol) concentrations were all <50 ng/ml...”

Line 279: Changed “give” to “obtain”

Line 281: Comment about using 20 ng/ml in place of <20 ng/ml: Cannot really do this

Comment to reviewer: We agree that this is not the best way to perform statistics. Unfortunately, if we do not keep this number in, our sample size becomes too small and we lose significance. With that in mind, we assigned it the highest value we possibly could knowing that it is likely lower than that value (which would better support our data). Even with it as the highest number that it could be, we were still able to show significance between the two groups.

Line 282: changed “read” to “analyzed”

Line 282: did not add “(retinol)” here as it seems repetitive given the change that was made to Line 277

Line 282: Change “vitamin A was” to “vitamin A concentrations were”

Table 3: Comment- Perhaps need to call your “groups” “treatments” throughout the MS

Comment to reviewer: The authors believe that the word “group” is less confusing to people overall and would prefer to keep it as is.

Line 295: Changed “between” to “among”

Line 297: Did not change “group” to “treatment” as was mentioned above.

Line 298: Changed “at” to “during” 

Line 301: Comment- This is not valid. Really better to exclude.

Comments to Reviewer: Omitted the two values for calcium and albumin as their removal did not change our p values and did not significantly change the mean, SD, or F statistics. However, in this particular instance, we still believe that they can be used as adequate place holders. For calcium- we were seeing that values decreased over time. By changing the >16 mg/dL baseline value to 16, we are actually shortchanging ourselves. For albumin- we were seeing decreases by day 35. By changing a less than 1 value to 1, we are again shortchanging ourselves of potentially an even greater change in values, but again still able to see a significant difference. If the rounding had changed values in the opposite directions from how things were trending, then we would definitely had removed them from the first submission.

Line 307: Comment- Title for Table 4 needs to be more comprehensive for table to stand alone.

Comment to reviewer: Title has been changed to “Average biochemistry results from all geckos over time (baseline vs. Day 35)”

Line 314: Added word “hepatic”

Line 316: Comment- What is the baseline vitamin A content of your BSFL?

 Comment to Reviewer: See line 249: 23 ± 30 µg/kg (as fed or 84 ± 109 µg/kg DMB)

Line 331: Changed “whole” to “intact”

Line 332-334: Comment- This is not a sentence- rewrite.

Comment to reviewer: “Protein, magnesium, potassium, sodium, iron, zinc, copper, and molybdenum also saw significant gains in digestibility (Table 2).”

Line 340: Changed “between” to “among”

Line 342-343: Changed “but this led to decreased…” to “but this can lead to decreased…”

Line 343-344: Added “(personal observation).”

Line 346: Added “when fed intact BSF larvae.” 

Line 347: Added “gecko mastication”

Line 346-348: Comment- Probably needs rephrased. The calcium is not “released” per se, but that likely that stomach acid has more surface area to act upon it if in smaller particles (i.e. ground).

Comment to reviewer: We agree, just difficult to say succinctly! How is this? “It would appear that gecko mastication or needle piercing does not provide enough disruption to the exoskeletal matrix to allow for calcium carbonate digestion.”

Line 350: Added “(personal observation).”

Line 368: Comment- In all treatment groups? Or which?

Comment to reviewer: Yes, all treatment groups saw a decline over time. Added “… for all treatment groups.”

Line 376: Changed “done” to “conducted” 

Line 379: Comment- With all diet treatments?

Comment to reviewer: Yes, all treatment groups saw a decline over. Added “… for all treatment groups.”

Line 380: Changed “drop” to “decrease”

Line 391: Removed the word “on”

Line 393: Removed “…consuming either whole BSF larvae or as the main ingredient…”

Line 399: Changed “Proving the second hypothesis regarding…” to “The second objective regarding…”

Line 400: Removed “a little”

Line 407-410: Changed to “Without baseline hepatic concentrations of vitamin A for individual geckos, it is impossible to truly evaluate a change in vitamin A status, but random assignment of the geckos to treatment groups should have minimized any bias.” 

Line 411: Removed “per MSU”

Line 412: Did not accept change from “unless the study became non-survival” to “unless the study design were altered” because we really want to highlight the fact that we were able to perform survival surgeries. Keeping the wording as is, is important to us here.

Line 413: Changed “sample size population” to “sample population sizes” 

Line 416: Added “concentrations”

Line 421-422: Comment- With 8 animals per treatment, could you have pooled at least some each sampling time to get longitudinal samples?

Comment to Reviewer: We could have, had we known at the beginning that their values were going to be low enough to require the higher sampling volumes. But we had already turned in our full samples to MSU and ended up not getting useable results back. We tried to use an ELISA plate assay for retinol during the middle part of the study (required less volume), but our results were very inconsistent with that method and we opted to leave those results out of the manuscript (hence why there is a big gap between blood draws in the middle of the study).

Line 423: Added “(measured as retinol)”

Line 438-439: Changed to “Group 1 and 2 were combined, providing a single vitamin A gut loaded treatment group versus the control (Group 3).” We did not use the phrase “value” as suggested because it made sound like there was only one value (n=1) for all of the treatment group (instead of n=8).

Line 441-442: Change to “Liver vitamin A concentrations also differed significantly between the vitamin A gut loaded and control dietary treatment groups.”

Line 443: Comment- redundant; This line has been omitted.

Line 443-448: Comment- Interesting but see if you can condense. 

Comment to reviewer: “Across both treatment groups, liver vitamin A concentrations ranged from 2.9-77.98 µg/g, with only a few individuals near the upper end of this range. These geckos likely had higher liver concentrations at the start of the study compared to the others. Age, diet, and husbandry conditions prior to the study (all of which are unknown), are likely contributors to the wide range seen. Paired liver samples would have helped limit this variance, but due to the limitations already discussed, were not performed.”

Line 454: Added “fed” and changed “had” to “displayed”

Line 457-463: Comments- Already reported. Do not repeat but rather incorporate the comparison more concisely. Tighten up (whole section).

Comment to Reviewer: “The values from the present study are much higher, with differences in sex (all female vs. all male study designs) and age being major influencing variables. Females would potentially have lower body stores of vitamin A due to large quantities being stored in the developing eggs. Cojean’s geckos were also 6-9 months old compared to ours that were mostly thought to be adults (> 10 months old). Vitamin A tends to accumulate in the liver as an animal ages leading to potential differences in the two populations.”

Line 474: Added “pre-formed”

Line 476-477: Comment- Or other carotenoids—mixes have been shown more effective for some insectivorous species… Don’t discount that… and its probably more realistic.

Comment to reviewer: We added “should probably be used in combination along with other carotenoid sources” to the end of the paragraph.

Line 478-479: Added “pre-formed” and changed “beta-carotene” to “appropriate precursors”. Added “insectivorous” and changed “will eventually become at risk” to “are at a high risk”.

Line 486: Added “across all treatments and was not correlated with measured plasma or liver vitamin A concentrations.”

Line 488-505: Left this paragraph in as the authors wanted to highlight what lesions would be consistent with hypovitaminosis A.

Line 508-509: Changed “multi-factorial, with a main issue being palatability.”

Line 511-513: Changed “Whether food preferences were due to palatability or a lack of movement of BSF larvae compared to crickets was outside the scope of this study.”

Line 513-514: Changed “Another contributor to inappetence and weight loss was infection with Cryptosporidium sp.” 

Line 514-515: We did not accept addition of colon and merger of sentences. Makes the resulting sentence too long. We did shorten the second sentence with your other suggested changes though. Changed “At least three geckos were confirmed to be infected at necropsy, but none showed obvious signs of infection until after…” 

Line 516: Changed “having intermittent” to “mixed”.

Line 520: Removed two sentences.

Line 522: Removed one sentence.

Line 522: Changed to “The other possible health limitation found on necropsy was stomatitis; lesions were not seen grossly…”

Line 528: Added “intact”. Removed “much” and “was”.

Line 529-530: Added “Despite high nutrient digestibility for proximate constituents, calcium digestibility, remained low.”

Line 533-534: Added “such as vitamin A incorporated into feeder BSF larvae.”

Reviewer 2 comments: 

Overall: As detailed in my comments to the authors my main concern is the conclusion that BSFL should be supplemented with calcium before being fed to leopard geckos. The data as presented does not seem to support that conclusion. Hence why I replied partly to question #1.

Comment to reviewer: Thank you for pointing out your concerns. We do agree with you that theoretically it should be enough calcium to support physiological needs based on the estimated minimum requirements that we have reported for leopard geckos. But we do think that people should still be cautious, especially given the comment you made about not all BSF larvae having a positive calcium to phosphorous ratio. We have changed our comments around accordingly (see below for specifics). 

Line 28-29 & 34-36: I’m not sure the data supports these statements. See comment #9 for more details why this might be.

Comment to reviewer: For the abstract we changed the lines to read “… with the exception of calcium (digestibility co-efficient 43%), as the calcium-rich exoskeleton usually remained intact after passage through the GI tract” and “While leopard geckos are able to digest most of the nutrients provided by BSF larvae, including those that have been gut loaded, more research needs to be performed to assess whether or not they provide adequate calcium in their non-supplemented form.”

Line 42-44: While 3 years ago I would have agreed with this statement there is now data showing the calcium content of BSFL is highly variable (more so than I ever would have guessed) and diet dependent. A review of 3 articles (shown below) shows calcium levels from 0.12 (similar to that for other insects) to 6.66% (dry matter basis) and Ca:P ranging from 0.30:1 to 14.9:1. It would be good for the authors to mention this somewhere so veterinarians and zoo nutritionists understand that the diet plays a critical role in BSFL calcium content and that live soldier fly larvae sold commercially can varying widely in calcium content. 

Spranghers T., Ottoboni M., Klootwijk C., Ovyn A., Deboosere S., De Meulenaer B., Michiels J., Eeckhout M., De Clercq P De Smet S., 2017. Nutritional composition of black soldier fly (Hermetia illucens) prepupae reared on different organic waste substrates. Journal of the Science of Food and Agriculture 97:2594-2600.

Tschirner M., and Simon A., 2015. Influence of different growing substrates and processing on the nutrient composition of black soldier fly larvae destined for animal feed. Journal of Insects as Food and Feed 1:249-259.

Wang S.Y., Wu L., Li B., Zhang, D. 2019. Reproductive potential and nutritional composition of Hermetia illucens (Diptera: Stratiomyidae) prepupae reared on different organic wastes. Journal of Economic Entomology doi: 10.1093/jee/toz296

Comment to reviewer: Thank you for this information! We have added these sources and changed the line slightly. We also addressed this subject more fully in the discussion. 

“…they are the only commercially produced insect that has been found to potentially have a natural positive calcium to phosphorous (Ca:P) ratio (2.5:1) based on their diet.”

In discussion, line 362-364: “Additionally, calcium content of BSF larvae can vary greatly based on their rearing diet’s composition. Recent studies have shown calcium:phosphorous ratios can range anywhere between 0.3:1 to 14.9:1 [4-5].”

Line 65: There is now data available to show that insects can be an excellent source of vitamin D if exposed to UV light which might help explain how nocturnal insectivores get their vitamin D. That article is Oonincx DGAB, van Keulen P, Finke MD, Baines FM, Vermeulen M, Bosch G. 2018. Evidence of vitamin D synthesis in insects exposed to UVb light. Nature - Scientific Reports DOI:10.1038/s41598-018-29232-w

Comment to reviewer: This information has been added.

“…however, recent research has found that some insects can produce vitamins D2 and D3 secondary to ultraviolet B radiation exposure, similar to vertebrates [14].”

Line 222-226: It might be nice to tell the reader the breakout of the 10 geckos that lost weight and the 7 that were removed (i.e. how many came from each of the three groups)

Comment to reviewer: Each breakout has been provided. The breakout for the seven necropsied ones are listed in two separate places now though. If this is a concern, please let us know which one to remove.

Line 195: “Seven geckos (Group 1, n=3; Group 2, n=2; Group 3, n=2) were euthanized intra-operatively after collection of the biopsies…”. This was the original placement.

Line 222-226: “Over the course of the experiment, 10 (41.6%) out of 24 leopard geckos lost weight (mean ± SD: -13.76 ± 7.71% weight loss, range: -0.14% to -23.91%)(Group 1, n=4; Group 2, n=3; Group 3, n=3). Seven (29.2%) of these geckos experienced inappetence and weight loss severe enough to require early removal from the study (Group 1, n=3; Group 2, n=2; Group 3, n=2).” These were the ones that were added per reviewer suggestions.

Line 243-247: I wonder if the authors might graph wt change by food intake (either % of body wt or perhaps ME) and they might be able to generate a nice regression line showing the maintenance energy requirements of these geckos. Might be a nice figure to add if there is a good correlation between intake and weight change.

Comment to reviewer: We looked at this information per your request but there was not a good correlation. Example: I had two geckos that ate approximately 80% of the food I offered them (food amounts were based on 5%BW) and one lost over 8% of his weight over time and the other gained 20% of his starting weight over time. Almost all of them did weird things like this.

Table 2: Perhaps here or somewhere else it might be a good idea to show the readers the average nutrient composition of the BSFL used in this experiment. I realize the raw data is in the supplemental appendix but perhaps to help the reader the average data (at least moisture, protein, fat, ash, Ca and P) might be shown in the article.

 Comment to reviewer: Added the requested nutrient composition to Table 1 (column 1).

Table 3: Does the statistical significance change for the plasma vit A data if the sample that was reported as <20 is assigned a value of 0 rather than 20? Seems unusual to assign it a 20.

Comment to reviewer: The value that was reported as <20 ng/ml was in the non-gut loaded group. Assigning it as a zero would have looked as if we were trying to skew the numbers to show that plasma vitamin A was lower in that group, so we assigned it the highest value we could which was a 20. Even with it as the highest number that it could be we were still able to show significance between the two groups, which was enough for us. If we take the value out completely, our results lose significance due to a smaller sample size which is why we decided to keep it in (p=0.04 vs 0.09).

Table 3: Does the statistical significance change for the liver vit A data if the outlier sample of 61.35 ug/g is included in the non-gut loaded group?

Comment to reviewer: Yes it does, p=0.03 (with the outlier removed) vs. p=0.15 (with it left in). Since we were able to follow the statistical rules regarding outlier removal, we felt comfortable removing it.

Line 352-366: I think this section needs to be reworded since the data does not appear to be consistent with previous statements (see comment #1). The authors estimate that “when adjusted for digestibility the larvae supplied 9.2 g Ca/kg diet” which is 8-51% HIGHER than the estimated calcium requirement for growing leopard geckos that they cite (6.1-8.5 g Ca/kg diet). Add to this the fact that these geckos were adults likely means the calcium intake was likely 200-300% of the adult requirement. It is well known that calcium regulation in most species occurs in the intestine. To quote the NRC Mineral Tolerance of Domestic Animals (page 99) “During positive calcium balance intestinal mechanism for absorption are shut down in most species”. That would appear to be the case here as even with relatively low absorption the geckos were likely consuming 2-3 times their requirement. So while low digestibility of calcium in the exoskeleton is certainly a possible explanation realistically feedback mechanisms regulating calcium absorption likely play as much if not a bigger role in the low calcium digestibility observed here.

Comment to reviewer: Thank you for the comment and we have adjusted the manuscript and our final conclusions to hopefully be a little less matter of fact regarding the larvae needing further supplementation given the points you make about adult geckos likely needing less calcium than growing juveniles. That is a very good point and we are glad you brought it up. However, I am not completely sold on the fact that the low digestibility is due to feedback mechanisms regulating the calcium absorption. Our reasoning for this goes back to Dierenfeld’s paper where she measured digestibility of dusted crickets as well. The frogs were able to digest >80% of that calcium despite a higher starting level compared to the BSF. While not a great comparison given the differences between our studies (different species, durations, etc) it is still enough to make me question the digestibility of the exoskeleton. Also, we had the personal observation of large exoskeleton pieces passing directly through the GI tract. Feedback mechanisms could definitely play a role in regulation of unbound calcium being digested, but I don’t think they would prevent the exoskeleton itself form breaking down. Here is our updated paragraph for this section:

“The estimated minimum dietary calcium requirement for growing leopard geckos is between 6.1 and 8.5 g Ca/kg diet (DMB) [18]. When adjusted for digestibility, BSF larvae in our study provided 9.2 g Ca/kg diet (DMB) which should be adequate to support calcium needs. It is possible that the calcium digestibility was low in these leopard geckos because they were adults and calcium absorption was impacted by normal feedback mechanisms. A study in fast-growing juvenile geckos would be helpful in further discerning whether low digestibility is a function of the BSF exoskeleton, as it would be expected to be higher in growing animals with a higher calcium requirement. 

However, the authors would still recommend caution due to the fact that physiological needs may vary based on species, age, reproductive status, and/or vitamin D status of the animal. Additionally, calcium content of BSF larvae can vary greatly based on their rearing diet’s composition. Recent studies have shown calcium:phosphorous ratios can range anywhere between 0.3:1 to 14.9:1 [4-5]. Thus, more research would be needed to prove that non-supplemented BSF larvae can, in fact, provide enough calcium to insectivorous reptiles based on how the insects are reared.”

Line 344-345(?): The statement “The increased phosphorus content would require a subsequent increase in calcium level to maintain the proper calcium to phosphorus level” seems overstated given how little data there is on optimum calcium-phosphorus ratios in reptiles and even for most vertebrates studied the ranges are fairly broad, typically 1:1 to 2:1. Suggest changing this to “The increased phosphorus content would require a subsequent increase in calcium level to maintain the SAME (emphasis mine) calcium to phosphorus level”

Comment to reviewer: We took this line out based on the changes to the paragraph we made for the previous comment.

Line 374-377: See the previous comment about calcium regulation (comment 9). As such a blanket recommendation regarding adding additional calcium to soldier fly larvae would seem to put animals at risk for excessive calcium intake which has been shown to inhibit the absorption of trace minerals putting the animal at risk for secondary trace mineral deficiencies. Add to that is the fact that the calcium content of black soldier fly larvae is highly variable and there is no standardized data for black soldier fly larvae sold commercially. As such I think this statement needs to either be removed or extensively modified.

Comment to reviewer: We changed this statement to be much less matter of fact. It now reads “At this time, we do not have enough data to recommend whether calcium supplementation is needed when offering BSF larvae to reptiles. Additional research needs to be conducted to establish true calcium requirements for reptiles and varied diets should always be offered to insectivores to limit the incidence of nutritional deficiencies.”

Line 422-428: Ploog reports plasma vitamin A values for Mississippi gopher frogs and while not a reptile it is an insectivore. Ploog C, Clunston R, Morris C, Iske C, Blanner W, Pessier A. 2015. Hypovitaminosis A: influence of three diets or topical treatment on hepatic, adipose, and plasma retinoid concentrations and presence of squamous metaplasia in Mississippi gopher frogs (Rana capito servosa). In Bissell H, Brooks M Eds. Proceedings of the Eleventh Conference on Zoo and Wildlife Nutrition, AZA Nutrition Advisory Group, Portland

Comment to reviewer: Thank you for the comment. We originally had values for reptiles and amphibians, but removed the amphibians for concision. Since none of the reptiles mentioned are insectivorous, we have added a few of them back and removed the chelonians. Now we can focus just on comparisons between squamates and insectivorous amphibians. Their plasma values still fall lower than most. 

“Previous literature has reported plasma vitamin A (measured as retinol) concentrations for various squamates and amphibian species, including green iguanas (Iguana iguana, 52-75 ng/ml), eastern indigo snakes (Drymarchon couperi, 9 ng/ml), anacondas (Eunectes murinus, 80 ng/ml), Mississippi gopher frogs (Rana capito servosa, 36-43 ng/ml), marine toads (Bufo marinus, 60 ng/ml), Cuban tree frogs (Osteopilus septentrionalis 83 ng/ml), and Puerto Rican crested toads (Bufo lemur, 130 ng/ml)[25-27, 30-33].

Line 455-456: There is now data showing carnivores can convert beta-carotene to retinol although whether they can make sufficient retinol from beta-carotene to meet their needs is unclear (see Green, Tang, Lango, Klasing, & Fascetti. Domestic cats convert [2H8]-β-carotene to [2H4]-retinol following a single oral dose. J Anim Physiol Anim Nutr 96:681-92.) Since in some studies that show no conversion the animals were fed or were previously fed diets containing high levels of retinol (often from liver) it may be that feedback mechanisms designed to prevent vitamin A toxicity were in play. Note Mississippi gopher frogs appear to be able to convert carotenoids to retinol. See Ploog at al cited in the previous comment.

Comment to reviewer: Thank you for the comment. We did not end up changing this sentence since there still there is still much more to study regarding this aspect in reptiles.

Line 493-496: Agree however the amount of time required for depletion likely depends on vitamin A stores in the liver so depending on how the animals are fed it may take less than or much more than 6 months to show signs of deficiency.

Comment to reviewer: Changed sentence to “Research in humans and other adult vertebrates confirm that >6 months of depletion is usually needed before clinical signs of hypovitaminosis A are detectable, but will obviously depend on the vitamin A status of the individual prior to the start of depletion [36-38].”

Line 530-533: I think these two sentences need to be modified based on the data shown and my comments (#9).

Comment to reviewer: “While calcium levels were likely adequate for leopard geckos based on estimated calcium requirements for the species, further research is needed to verify this assumption and to determine calcium requirements for other insectivorous species.”

Reviewer 3 comments:

Overall: this is a well written manuscript and a well designed and important study. Some minor comments listed below.

Line 33-34: report p values as 0.04 and 0.03. This change has been made.

Line 78: Please provide a reference for this statement (leopard geckos prone to vit a deficiency, etc).

References have been added to this sentence.

Line 109: provide reference that leopard geckos develop ocular changes due to vitamin A deficiency

The reference has been added.

Line 195, 224, 226, 236: the 7 emaciated geckos were from all 3 treatments groups? Would be worth reporting here.

 These lines have all been updated to show distribution across groups.

Line 260-262: the statement regarding the frogs is inappropriate for the result section and should be moved to the discussion section.

Comment to reviewer: Usually we would agree with you. However, we really wanted the two digestibility tables to be placed next to each in the manuscript (and not interrupted by having the biochemistry table and vitamin A table in between). We found it easier for the purposes of flow to place it here as we were already describing how the first column represented the three treatment groups being averaged together. We also performed statistics between our data and the frog data to determine if there were any significant differences between what we reported and what was already published. If you would still like it moved, we absolutely can. These were just the reasons why we preferred it here.

Line 298: possible that this animal was a female, and not a male?

This gecko was definitely male based on external characteristics. This does not rule out the possibility of an ovotestis, which has been reported in various lizard species. Since we have now removed the value from statistical analysis, we do not believe that it needs to be discussed as a possibility in the manuscript.

Line 333-338: were the larvae fed in your study and in the frog study of same size/age? Possible that older/larger larvae are harder to digest or have different nurtritional values than smaller/younger ones?

Comment to reviewer: Thank you for the comment. This is definitely possible. In looking back at the previous research, the frog study used medium sized larvae, whereas we used large sized larvae (weighed twice as much on average). I also looked at the frog study’s nutrient composition versus our own. DM and crude protein were very similar between the two studies. However, the frogs’ mineral levels were all moderately higher meaning that it is possible that we saw higher digestibilities for those nutrients because there was less to begin with. We have added a comment to bring up this point.

“Some of the differences in digestibility could be related to differences in larval composition between the studies. However, given that most of our digestibility coefficients were more similar to the values reported for mashed larvae from their study rather than the values for intact larvae, we believe that most of the differences were due to a higher degree of mastication by the leopard geckos which would allow for digestive enzymes to breach the tough exoskeleton and breakdown the inner portions of the larvae.”

Line 350-359: Or it could be due to a vitamin D deficiency. What was the vit D3 content of the larvae. Did they receive enough vit D3? Since no UVB source was provided, dietary vit D3 content of the offered diet should be discussed here as a potential cause for lower Ca levels on day 35….

Comment to reviewer: Very astute point. We purposefully did not supplement with vitamin D3 or UVB because of its role in calcium absorption and it would have added another variable that we weren’t able to control very well. However, we did neglect to mention that vitamin D deficiency could be a cause of lower calcium levels. We have added a quick mention to two different spots.

Line 360-361: “However, the authors would still recommend caution due to the fact that physiological needs may vary based on species, age, reproductive status, and/or vitamin D status of the animal. “

Line 368-369: “Poor calcium digestibility or low levels of vitamin D could be possible causes for this decline.”

---

## [Decision Letter · Decision Letter 1]

16 Apr 2020

Digestibility of black soldier fly larvae (Hermetia illucens) fed to leopard geckos (Eublepharis macularius)

PONE-D-19-33399R1

Dear Dr. Boykin,

We are pleased to inform you that your manuscript has been judged scientifically suitable for publication and will be formally accepted for publication once it complies with all outstanding technical requirements.

With kind regards,

Jake Kerby, Ph.D.

Academic Editor

PLOS ONE

Additional Editor Comments (optional):

Thanks for addressing all the comments of the reviewers.

Reviewers' comments:

Reviewer's Responses to Questions

**Comments to the Author**

1. If the authors have adequately addressed your comments raised in a previous round of review and you feel that this manuscript is now acceptable for publication, you may indicate that here to bypass the “Comments to the Author” section, enter your conflict of interest statement in the “Confidential to Editor” section, and submit your "Accept" recommendation.

Reviewer #1: All comments have been addressed

Reviewer #2: All comments have been addressed

2. Is the manuscript technically sound, and do the data support the conclusions?

Reviewer #1: Partly

Reviewer #2: Yes

3. Has the statistical analysis been performed appropriately and rigorously? 

Reviewer #1: Yes

Reviewer #2: Yes

4. Have the authors made all data underlying the findings in their manuscript fully available?

Reviewer #1: Yes

Reviewer #2: Yes

5. Is the manuscript presented in an intelligible fashion and written in standard English?

Reviewer #1: Yes

Reviewer #2: Yes

6. Review Comments to the Author

Reviewer #1: Good job of addressing reviewer concerns and comments; statistics on small sample sizes remain problematic (and likely will with most exotics), and the assignment of values below detection limits is still not statistically appropriate, but your justification is acceptable.

Footnote for Table 1 should read "No significant differences....." (rather than no significance)

Reviewer #2: See attached for some minor comment but I liked the original manuscript and this revision makes it even better. Kudos to the authors for a job well done and contributing valuable information to the field.

7. PLOS authors have the option to publish the peer review history of their article (what does this mean?). If published, this will include your full peer review and any attached files.

Reviewer #1: No

Reviewer #2: No

---

## [Editor Report · Acceptance letter]

29 Apr 2020

PONE-D-19-33399R1 

Digestibility of black soldier fly larvae (*Hermetia illucens*) fed to leopard geckos (*Eublepharis macularius*) 

Dear Dr. Boykin:

I am pleased to inform you that your manuscript has been deemed suitable for publication in PLOS ONE. Congratulations! Your manuscript is now with our production department. 

With kind regards,

on behalf of

Dr. Jake Kerby 

Academic Editor

PLOS ONE